# Polyphyletic screen defines distinct classes of plant-derived natural products that oppose tauopathy

Chatrawee D Shepard[1,2] , Xinmin Chang[3], Paul M Seidler[3], Sean P Curran[2]

Alzheimer's disease (AD) is a debilitating neurodegenerative disease hallmarked by the presence of amyloid-$\beta$ (A$\beta$) plaques and tau fibrils but with limited treatment options. Here, we describe two plant-derived natural products with distinct mechanisms of action on tau fibril disaggregation and prionogenic seeding. We first characterized the effects of oolonghomobisflavan A (OFA) and oolonghomobisflavan B (OFB) treatments, which alter the transcriptional landscape toward enhanced proteostasis and reverse the shortened lifespan in a *Caenorhabditis elegans* model expressing pathogenic human tau ("hTau-expressing"), but increase healthspan. Mechanistically, OFA treatment reduced the burden of tau protein aggregation and promoted tau disaggregation in hTau-expressing *C. elegans* and also inhibited seeding in assays using ex vivo brain-derived paired helical filament tau protein fibrils from Alzheimer's disease brain donors. We leveraged this finding to develop a facile screening approach for compounds, like OFA, that could mitigate tau aggregation, which revealed plumbagin (PB) as a potent inhibitor of tau monomer aggregation which is mechanistically distinct from the tau fibril disaggregase activity associated with OFA. Collectively, this study reveals a new strategy for identifying therapeutic compounds that target tauopathy and provides mechanistic insight supporting the neuroprotective effects of plant-derived natural products.

## Introduction

Age-associated neurodegenerative diseases are a major and growing public health challenge. Increased life expectancy and an absence of effective pharmacological treatments have left elderly, caregivers, and the healthcare system reeling (1). Alzheimer's disease (AD) is the most common neurodegenerative diseases characterized by cognition and memory impairments (1). AD histopathology involves neuro-inflammation and accumulation of amyloid $\beta$-protein (A$\beta$) and neurofibrillary tangles (NFTs), which drives broad neuronal dysfunction (1, 2) resulting in cognitive and motor deficits associated with neuronal cell death. Over 60% of all diagnosed cases of dementia are attributed to AD, and an estimated 40% of AD cases may be prevented or delayed by modifiable risk factors including diet, midlife obesity, and diabetes (3). Significant focus on the role amyloid plays in AD has led to FDA-approved cholinergic antagonists and anti-amyloid antibodies as treatments of AD, but these have not demonstrably slowed cognitive decline (albeit the full lecanemab and other new clinical trial data are pending (4, 5, 6)). As such, new molecular targets must be identified to advance the AD therapeutic discovery pipeline (7).

Tau is a microtubule-associated protein that interacts with tubulin (2). Under physiological conditions, tau has been shown to catalyze microtubule assembly, which impacts axonal transport and the structural organization of the synapse (1, 2). In tauopathies and AD, pathological transformation of tau begins with hyperphosphorylation, conformational changes of protein structure, loss of microtubule binding, oligomerization, misfolding, and ultimately the formation of insoluble filaments that accumulate as NFTs (1, 2). Hyperphosphorylation of tau further promotes the formation of proteotoxic intracellular amyloid aggregates that impact neurodegenerative diseases (2). The loss of microtubule stability because of abnormal tau phosphorylation has been reported as a major cause of tauopathies (2).

*Caenorhabditis elegans* (*C. elegans*) has been extensively used as a model of neurodegenerative diseases (8). Transgenic *C. elegans* strain KAE112 (hereafter referred to as "hTau o/e") has been created with codon-optimized human 0N4R V337M tau expressed in the body wall muscle to better understand the impact of aggregation tau on pathophysiologic and cellular function (9). In this model, the hyperphosphorylation of the human tau variant drives proteotoxicity, resulting in premature defects in age-associated health metrics, including reproductive fitness, developmental rate, muscle paralysis, and lifespan (9). Increasing evidence suggests molecules that target tau aggregation could be useful in supporting healthier aging and possibly treating various stages of AD and other tauopathies involving tau aggregation (10). The *C. elegans* hTau-expressing model is a practical tool for discovering therapeutic molecules with anti-tau effects.

Several studies have reported oxidative stress and neuronal cell damage as key drivers of protein aggregation (11). Natural products from herbs or plant extracts with potent antioxidants that inhibit tau aggregation could provide an alternative approach to treat or

[1]The Appalachian Natural Products Research Program, Department of Biomedical Sciences, The Joan C. Edwards School of Medicine, Marshall University, Huntington, WV, USA [2]Leonard Davis School of Gerontology, University of Southern California, Los Angeles, CA, USA [3]Department of Pharmacology and Pharmaceutical Sciences, University of Southern California, Los Angeles, CA, USA

Correspondence: duangjan@marshall.edu; spcurran@usc.edu

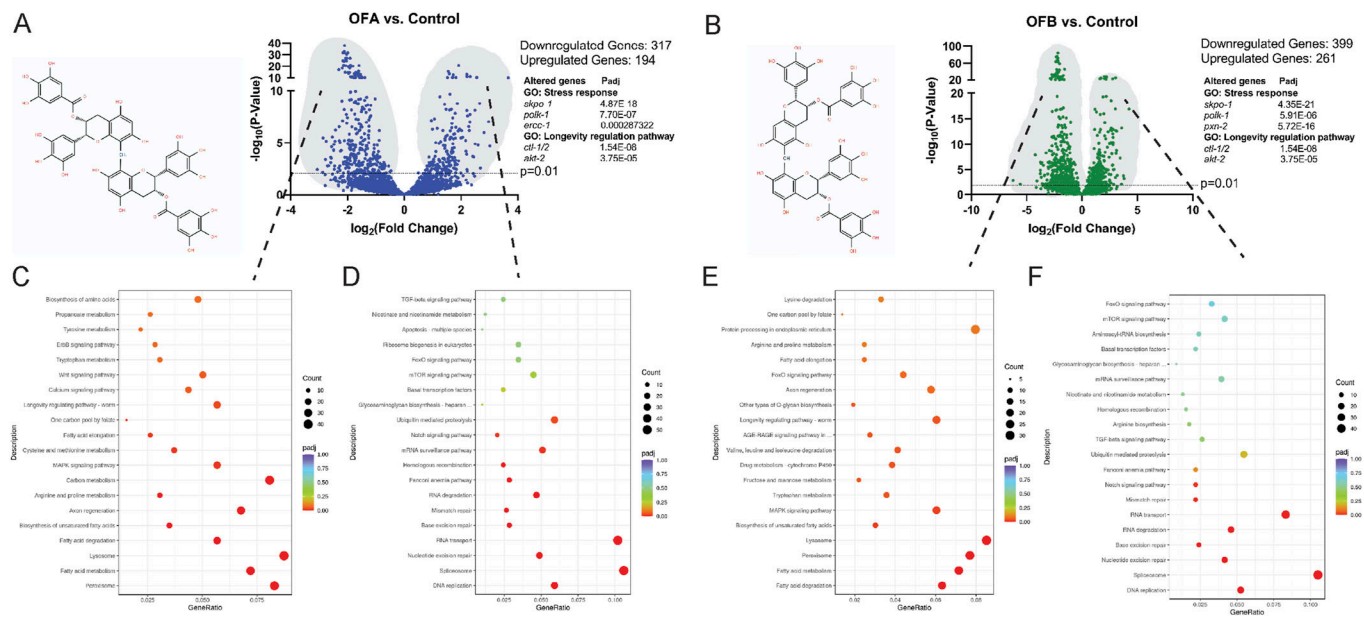

**Figure 1. OF treatment induces a healthspan-promoting transcriptional signature.**
**(A, B)** Volcano plots of differentially expressed genes between mock-treated controls and OFA (A) and OFB (B); top five representative genes with significant changes (see Table S1). **(C, D, E, F)** As compared to mock treatment (control), gene ontology (GO) and KEGG enrichment analysis of genes that decrease (C, E) and increase (D, F) expression in OFA and OFB treatment, respectively. The mean expression level for each gene is indicated by log₂FoldChange. All genes were considered to be significant with an adjusted *P*-value < 0.05. RNA was extracted from three independent biological replicates per condition.

prevent neurodegenerative diseases. Tea polyphenols act as natural bioactive compounds that could complement traditional therapeutic agents for neurodegenerative diseases characterized by proteostasis defects (12, 13), including Aβ (14, 15, 16), tau (17), α-synuclein (15), inflammation (18), and oxidative stress (14, 15, 16).

Oolong tea (*Camellia sinensis*) has been studied for beneficial effects on neurodegenerative diseases (18, 19), but the molecular mechanisms underlying the neuroprotective effects of bioactive compounds in oolong tea could be broad and require further investigation. In previous work, we identified the oxidative stress resistance properties and neuroprotective effects against Aβ of oolong tea extracts and its bioactive molecules oolonghomobisflavans (OFs) (14, 15, 16). In this study, we further identified the specific action of oolonghomobisflavan A (OFA) and oolonghomobisflavan B (OFB) on tau fibrils, aggregation, and tau protein–induced toxicity. Extrapolating from these results, we find that treating transgenic tau *C. elegans* models with a brain-permeable phenol elicits similar rescue effects. The molecules we identified retain inhibitory activity against prion-like seeding by human AD brain homogenates, suggesting potentially valuable molecules and strategies for aging-related proteinopathies through the development of new nutraceutical preparations.

# Results

## OFA and OFB treatments induce a proteostasis-enhancing and health-promoting transcriptional signature

Based on the known effects of OFA and OFB (OFs) on health and longevity effects in WT animals (14, 16) and amyloid models of proteinopathy (14), we wanted to understand the molecular basis of this response. To that end, we first compared the transcriptional profile of WT with and without OF treatment by RNA-seq. 511 mRNA transcripts are significantly different (including 194 up-regulated and 317 down-regulated) with OFA treatment when compared to untreated controls (Fig 1A and Table S1), 660 mRNA transcripts (including 261 up-regulated and 399 down-regulated transcripts) are significantly different in OFB treatment when compared to untreated controls (Fig 1B and Table S1), and 72 mRNA transcripts (including 69 up-regulated and 3 down-regulated transcripts) are significantly different in OFA treatment when compared to OFB treatment (Fig S1A and B and Table S1).

We performed a gene enrichment analysis (GEA) for each treatment group and identified classes of transcripts significantly regulated by OFs (Figs 1C–F and S1C–K, and Table S1). Several oxidative stress–related terms were identified, including response to stress (GO:0006950), cellular response to stress (GO:0033554), cellular response to DNA damage stimulus (GO:0006974), oxidation–reduction process (GO: 0055114), and oxidoreductase activity (GO:0016491). KEGG analysis revealed enrichment for longevity-regulating pathways (KO:04212 and KO:04213), including FoxO (KO:04068), mTOR (KO:04150), and MAPK (KO:04010) signaling pathways. Importantly, among the genes differentially altered between OFA and OFB treatments, the expression of several ubiquitin-mediated proteolysis (KO:04120 and GO:0016579), Rho and RAS signaling (GO:0017048 and GO:0017016), and axon regeneration (KO:04361) genes was regulated by OFA treatment, albeit near significance for OFB. Taken together, these results demonstrate that OF treatment influences the expression of genes affecting oxidative stress responses, and cellular proteostasis and

signaling. Unsurprisingly, based on their related chemical structures, effects of OFA and OFB treatments on the transcriptional landscape are remarkably similar. These data indicate that the pathway effects exerted by OFA and OFB are both robust and specific.

## OF treatment reverses physiological detriments of tauopathy in *C. elegans*

The ability of OFs to reduce tau proteinopathy both in vivo and in vitro suggested that OFs would also be able to alleviate the health-related detriments stemming from the expression of hTau. Because we previously noted a general improvement in health with age in animals treated with OFs, we next characterized the effects of OFs in detail in both WT animals and animals expressing pathogenic human tau variants (0N4R;V337M); strain "hTau o/e" (9, 20). This model expresses a codon-optimized human tau under the control of the muscle myosin (*myo-3*) promoter, which restricts expression primarily to the body wall muscle. The expression of this mutant tau variant is associated with reduced lifespan and diminished health (9).

Previous studies have documented the negative effects of tau proteotoxicity on multiple fitness parameters, developmental growth, and timing and brood size (9). OFA and OFB treatments were both capable of significantly reversing the impaired reproductive output of hTau-expressing worms (Figs 2A and B and S2A, and Table S2), specifically toward the end of the reproductive span at days 3–5 of adulthood (Fig S2A and Table S2). In contrast, OFA and OFB treatments had no effect on the slowed development and growth observed in hTau-expressing animals (Fig 2C and D).

We next measured the impact of OFA and OFB treatments on the decline in muscle function resulting from hTau expression (9). We confirmed the impaired crawling and thrashing speed in hTau-expressing animals and noted a significant improvement of both movement parameters with treatment of either OFA or OFB at day 2 and day 4 of adulthood. By day 4, thrashing speed showed significant improvement with treatment, whereas crawling speed did not exhibit significant changes (Figs 2E–H and S2B and C). Importantly, the progressive movement paralysis stemming from pathogenic hTau expression is significantly delayed, by ~4 d, in animals treated with OF as compared to the mock-treated control group (Fig 2I). Lastly, pharyngeal function is a facile biomarker of aging in *C. elegans* (21), which displays an accelerated rate of decline in hTau-expressing animals (Fig S2D). We previously demonstrated that OF treatment could protect pharyngeal function with age in WT animals (14), and similarly, pharyngeal pumping rate was significantly improved with OF treatment in hTau-expressing animals on days 5, 7, 9, and 11 of adulthood as compared to mock-treated controls (Figs 2J and S2D).

## OFs reverse the shortened lifespan associated with the *C. elegans* model of tauopathy

To complement our physiological assessments, we next conducted lifespan assays to evaluate the organism-level impact of OF treatment. We confirmed the shortened lifespan previously documented in animals expressing pathogenic human tau (Fig 3A and Table S3), but critically, we found that both WT and hTau-expressing (hTau o/e) worms treated with OFA and OFB at the L4 larva stage display a significant extension of lifespan (at 20°C) as compared to mock-treated controls (Fig 3B and C and Table S3). In general, cotreatment with both OFA and OFB (OFAB) did not provide any synergetic effects (Fig S3A and B and Table S3), which suggests OFA and OFB extend lifespan by similar mechanisms; also predicted by the remarkably similar transcriptional profiles we measured (Figs 1 and S1). Collectively, these data reveal that treatment with OFs can drive a lifespan-promoting enhancement of organismal health, but more importantly can significantly delay the age-related dysfunction in the context of hTau-related proteotoxicity.

## OFs disaggregate human tau fibrils in both in vivo and in situ models

In early stages of Alzheimer's disease, tau becomes hyper-phosphorylated and mislocalized, which can contribute to its aggregation and toxicity (22, 23), and this hyperphosphorylation is mimicked in the *C. elegans* hTau-expressing model (9). To measure the impact of OF treatment on tau aggregation and proteotoxicity, we first examined the phosphorylation status of tau as measured by the abundance of phosphorylated tau (pTau) on residues S202 and S416, which were reduced by ~81% to 82% and ~78% to 88%, respectively (Fig 4A–C). We also observed a significant reduction in total hTau protein with both OFA and OFA treatments (Fig 4D and E), but perhaps most importantly a reduction of aggregated species (immunoreactive species detected that migrate slower than the 50-kD monomer). Importantly, when pTau levels are normalized to total tau expression, we noted that the pTau/total tau ratio was significantly reduced at both phosphorylation sites, S202 (~78%) and S416 (~82%). In hTau-expressing (hTau o/e) worms, tau expression is driven by the *myo-3* promoter. RNA-seq data from OF-treated WT animals revealed down-regulation of *myo-3* (K12F2.1) (Table S1), suggesting that reduced tau levels in hTau-expressing (hTau o/e) worms may partially be due to a decrease in transgene expression. Consistently, total tau protein levels reduced after treatment (Fig 4D and E); however, the pronounced reduction in the pTau/total tau ratio indicates that OFA and OFB affect tau posttranslationally, potentially by interfering with phosphorylation or aggregation pathways. Taken together, these results reveal that OFA and OFB influence tau proteostasis.

Tau aggregation results in the formation of proteotoxic fibrils that propagate and drive neurodegeneration by prion-like seeding. Through the seeding mechanism, tau aggregation spreads from one cell to others (2). We next examined whether OF treatment could inhibit tau fibril formation in a biosensor cell assay that measures fibril propagation by seeding. Tau biosensor models have been used to identify tau inhibitors with fibril disaggregases (17, 24 *Preprint*). We used crude AD patient–derived brain homogenates as the seed in a tau biosensor cell assay, which revealed a dose-dependent inhibition of seeding by both OFA and OFB treatment groups, as compared to mock-treated controls (Fig 4F–J).

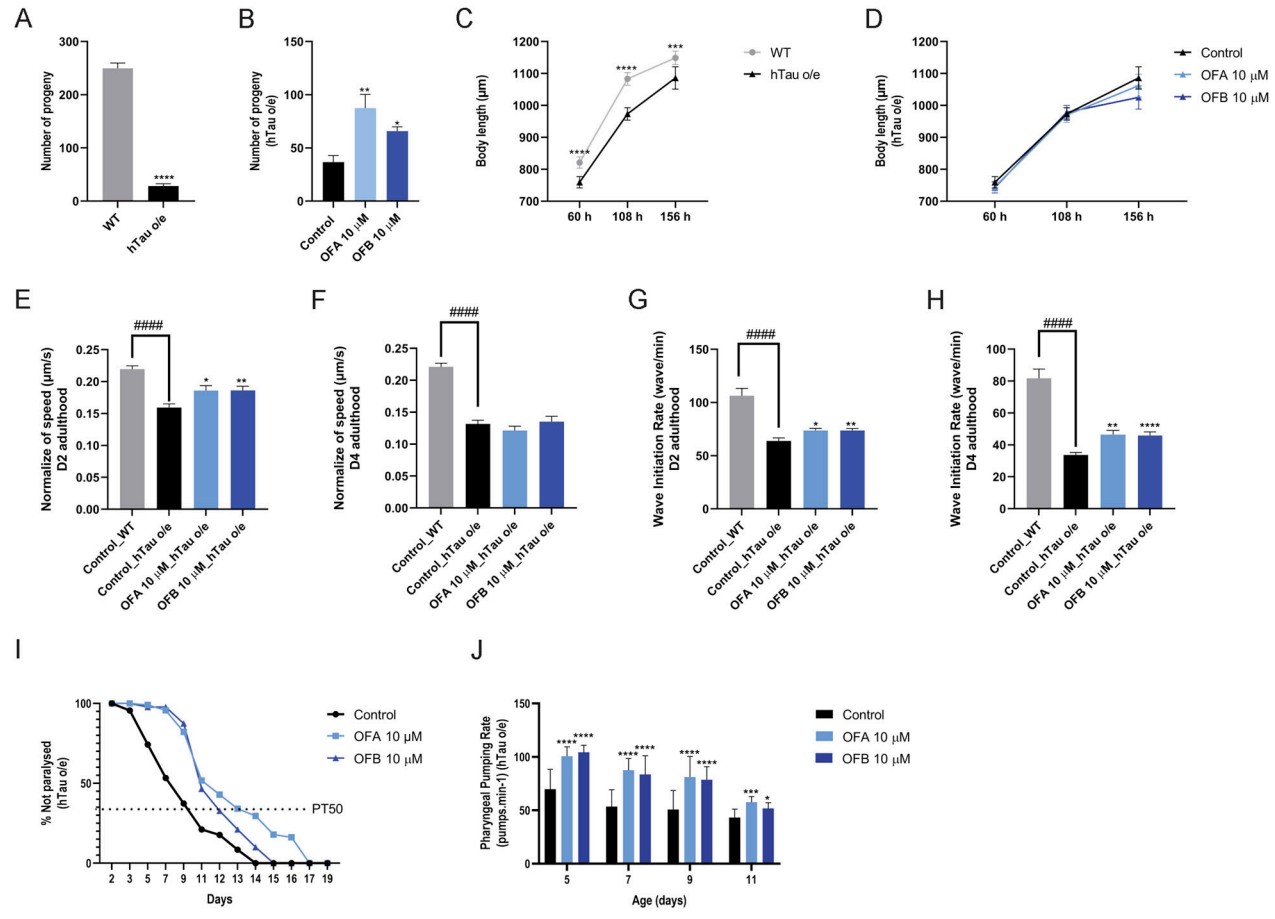

**Figure 2. Effects of OFs on health metrics of hTau-expressing animals.**
**(A, B, C, D)** Comparison of total viable progeny (A, B) and body size (C, D) between WT and hTau-expressing (hTau o/e) worms treated with OFA and OFB, as compared to mock treatment (control). **(E, F, G, H, I, J)** Effect of OFA and OFB on crawling speed (E, F), thrashing (G, H), movement paralysis (I), and pharyngeal pumping rate (J) in WT and hTau-expressing (hTau o/e) worms, as compared to mock treatment (control).*$P < 0.05$, **$P < 0.01$, ***$P < 0.001$, and ****$P < 0.0001$, compared with the mock-treated controls by one-way ANOVA followed by Bonferroni's method (post hoc). n ≥ 30; N = 3. Data represent the mean ± SEM from at least three independent biological replicates.

Quantitative electron microscopy (qEM) data shown in Fig 4K–N demonstrate that AD brain–derived tau fibrils that are incubated with OFA have a reduced propensity for aggregation. In a time course experiment over 24 h, as compared to the abundance of tau fibrils (red arrows) present at the start of the assay (0 h), by 3 h of incubation OFA condensates (white arrows) become more pronounced with instances of condensates coalescing with AD tau paired helical filament (PHF) fibrils (blue arrows) (Fig S4A). By 9 h, tau fibrils lose their fibril-like morphology, suggesting disaggregation of tau fibrils by OFA treatment (Figs 4L and S4B). At 24 h, the presence of tau fibrils is significantly reduced, and fibrils that remain are largely encapsulated by OFA condensates (blue arrows) (Figs 4M and S4C). Images quantified after 24 h of treatment with OFA show a ~95% reduction in AD tau PHFs (Fig 4O). These data align with findings from cellular and physiological models, which demonstrate the inhibitory effects of oolonghomobisflavans in mitigating tau fibril pathology (Figs 2, 3, and 4).

## Discovery of cell- and brain-permeable phenols with anti-tau effects in *C. elegans* models

We leveraged the facile hTau-expressing *C. elegans* model to screen additional compounds for potential anti-tau aggregation effects with the goal of identifying molecules that would inhibit hTau proteopathy (Fig 5). We hypothesized that molecules with similar chemical properties to OFA, which can increase lifespan and improve reproductive output in hTau-expressing animals, might be potent disaggregases for tau fibrils. Therefore, we screened a handful of phenols using the hTau-expressing (hTau o/e) *C. elegans* model to identify candidates for screening in tau fibril inhibitor assays.

We performed screening with several polyphenols with documented antioxidative effects. These included ascorbic acid (VC), caffeic acid (CA), echinatin (EC), N-acetylcysteine (NAC), and plumbagin (PB) (Figs 5A–D and S5A–N). PB, a natural product derived from *Plumbago zeylanica*, significantly increased the lifespan of both hTau-expressing (hTau o/e) and WT strains (Fig 5A

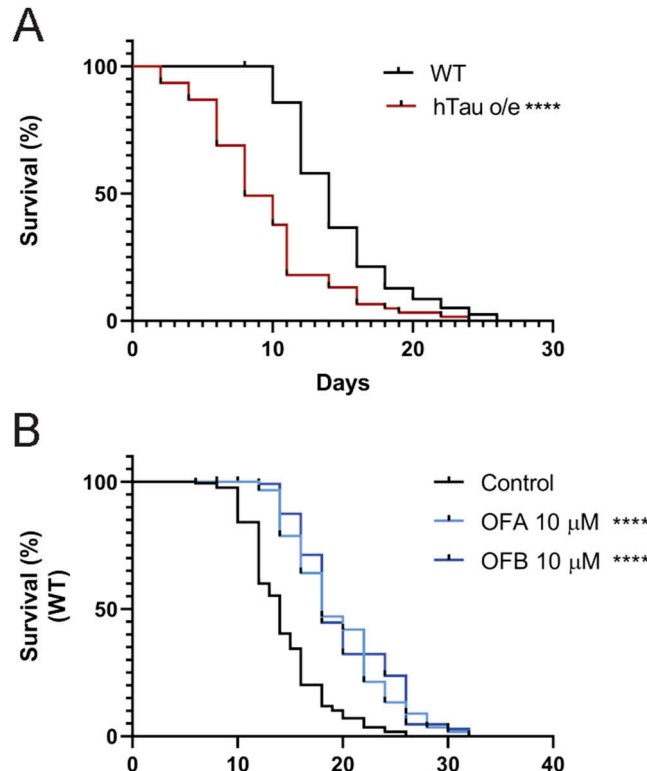

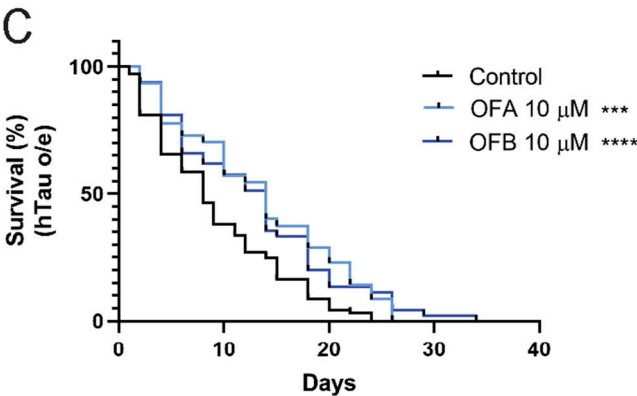

**Figure 3. Effects of OFs on lifespan of hTau-expressing worms.**
**(A)** Comparison of lifespan between WT and hTau-expressing (hTau o/e) worms (A). **(B, C)** Survival curves of WT (B) and hTau-expressing (hTau o/e) (C) worms treated with OFA and OFB. *$P < 0.05$, **$P < 0.01$, ***$P < 0.001$, and ****$P < 0.0001$, compared with the mock-treated controls by one-way ANOVA followed by a log-rank test. Worms were maintained at 20°C. n ≥ 30; N = 3. Data represent the mean ± SEM from at least three independent biological replicates.

and B), although VC, CA, EC, and NAC had no effect on lifespan (Fig S5G and H). Although PB significantly extended median lifespan in hTau-expressing (hTau o/e) worms (Fig 5B), it did not increase maximum lifespan or restore the early mortality-free plateau in survival curves (Fig 5B). This suggests PB partially mitigates tau-induced toxicity, primarily reducing midlife mortality without preventing early effects. Moreover, PB and NAC also increased reproductive output in the hTau-expressing *C. elegans* model

(Fig 5C and D), whereas VC, CA, and EC could not (Fig S5I, M, and N). None of the compounds influenced reproductive output in WT animals (Fig S5N). Because PB was the only one among the series tested that increased lifespan and reproductive output, we extended our experiments to investigate the effects of PB on tau aggregation using in vitro assays.

Strikingly, PB demonstrated a dose-dependent inhibition of seeding activity by the crude AD brain homogenate in tau K18 biosensor cell assays (Fig 6A). PB exhibited similar potency to OFA and OFB, with an IC50 of 7 (6.743) μM, compared with 3 (2.652) μM and 1 (1.318) μM for OFA and OFB, respectively (Fig S6A and B). In vitro seeding experiments (Fig 6B) using tau fibrils purified from the AD brain tissue confirm that the inhibited seeding is mediated by a direct, on-target mechanism, independent of potential cellular pathway effects. Therefore, we asked whether PB enhanced AD tau fibril disaggregation. Unlike OFs, qEM data in Fig 6C show that PB is not a tau fibril disaggregase. Rather, data in Fig 6D reveal that PB inhibits tau monomer aggregation. These findings indicate that the effects of PB involve a distinct mechanism of action compared with OFA and OFB.

## Discussion

The discovery of molecules that can treat AD has been challenging, but compounds that are able to interfere with the transformation process, which is catalyzed by prionogenic seeding by tau fibrils, could be a promising prophylactic or therapeutic strategy for neurodegenerative diseases. In this study, we illuminated a powerful approach for screening bioactive natural products derived from plants that can ameliorate tau aggregation. Compounds that reduce the physiological symptoms of overexpressing human pathogenic tau in *C. elegans* were then tested in a tau biosensor cell experiments using fibrils extracted from postmortem brains of AD patients. This approach identified two components that both act as potent inhibitors of tau pathology but remarkably act by different mechanisms.

We first examined oolonghomobisflavans (OFs), which are structurally distinct dimers of the tea polyphenol, epigallocatechin gallate (EGCG), that have emerged as potent antioxidant agents that can contribute to overall neuronal health (12, 17). Multiple lines of evidence suggest that EGCG has the capacity to impact tauopathy (17, 25, 26, 27), but the mechanistic details of these effects require additional investigation.

Although strikingly similar, some minor differences between the transcriptional profiles of OFA- and OFB-treated animals define the molecular underpinnings of this longevity response, particularly the enhancement of oxidative stress resistance, maintenance of DNA damage, and, perhaps most importantly, the enhancement of proteostasis and dissolution of tau aggregates that follows treatment with OFs. Molecules up-regulated by OFs that can contribute to the enhanced proteostasis include heat shock proteins (HSPs) that mitigate protein misfolding, aggregation, and accumulation (28), and several ubiquitin-mediated proteolysis genes including ubiquitin-specific proteases (29, 30, 31, 32). We observed tau aggregates at a variety of masses likely because of a

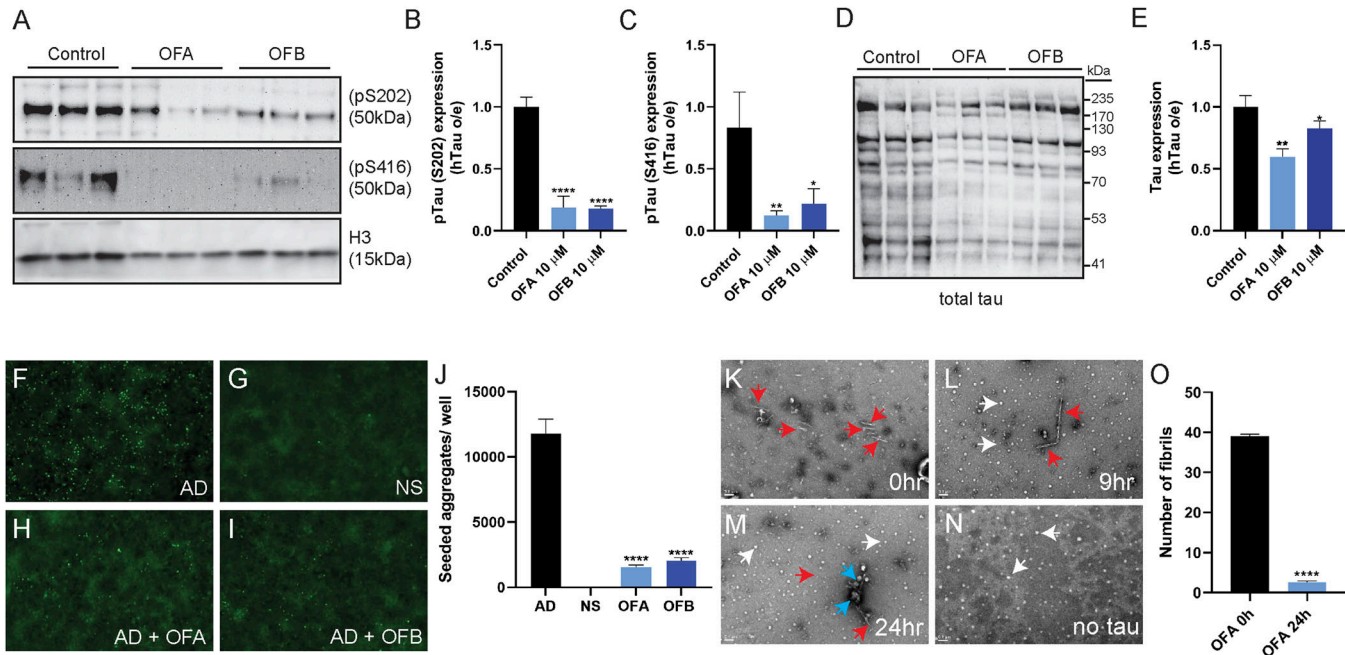

**Figure 4. OF treatment reduces tauopathy across model systems.**
**(A, B, C)** Western blot analysis of tau phosphorylation (S202, S416) in hTau-expressing (hTau o/e) worms after treatment with OFA and OFB relative to mock treatment (control) (A, B, C). **(D, E)** Western blot analysis of total tau protein (D) and quantification (E). n ≥ 30; N = 3. **(F, G, H, I, J)** Seeding inhibition measured by transfecting inhibitor-treated AD brain homogenate in fluorescent tau K18 biosensor cells (F, G, H, I, J); AD brain homogenate without added inhibitor (F), NS, no seed (G), OFA (H), and OFB (I). **(J)** Seeded aggregates were determined by quantifying the number of fluorescent puncta as a function of the indicated inhibitor and disaggregation (J). **(K, L, M, N, O)** Error bars represent SDs of triplicate measures. OFA-mediated AD tau fibril disaggregation, measured by qEM of AD tau fibrils (K, L, M, N, O). **(K, L, M, N)** Representative images shown of AD tau fibrils (red arrows), OFA condensates (white arrows), and OFA-associated fibrils (blue arrows) (K, L, M, N). **(O)** Fibrils quantified after 24-h incubation with OFA (O). Fibrils were counted from randomly acquired micrographs obtained by automated imaging using EPU software from N = 66 images. Fibril counts were obtained by splitting the image sets three ways. Error bars represent SDs. *P < 0.05, **P < 0.01, ***P < 0.001, and ****P < 0.0001, compared with the mock-treated controls by one-way ANOVA followed by Bonferroni's method (post hoc). Data represent the mean ± SEM from at least three independent biological replicates.

combination of biochemical properties including posttranslational modifications. Although the enhancement of the cellular proteostasis machinery is a likely contributor of the ability of OFs to maintain proteostasis even in the presence of pathogenic tau expression, our electron microscopy results suggest that OFA forms condensates, which functionally interact with and disaggregate tau fibrils. The aggregation of chemically similar polyphenols has been characterized at high resolution (33 *Preprint*) and is driven by phenol–phenol interactions. Condensates of OFA are suited to maintain productive phenol-mediated binding to tau fibrils given the excess number of aromatic hydroxyls (N = 16) for OFA and OFB.

Oxidative stress is closely related to age-related neurodegenerative diseases (1). Guided by these data, and our previous discovery that OFs can act as potent antioxidants, we leveraged *C. elegans* to rapidly screen a panel of antioxidant compounds. This facile screening model is based on our observed connection between the suppression of diminished reproduction and lifespan observed in the hTau-expressing worms and the capacity to disaggregate tau fibrils derived from AD patient brains.

We identified plumbagin (PB) as a cell- and brain-permeable flavonoid that improved the health of hTau-expressing worms similar to treatment of OFs. Although PB has been long known for antioxidant and neuroprotective effects, experimental evidence

only recently showed neuroprotective effects, with PB ameliorating memory dysfunction in streptozotocin models, which compromises hippocampal function and memory (34). PB was shown to be an inhibitor of MAPK4 (mitogen-activated protein kinase 4) (35), indicating that inhibiting tau phosphorylation is one potential mechanism of action (MOA) explaining neuroprotective effects of PB, although a dual role of antioxidant activity in neuroprotective effects cannot be dismissed. A potential limitation to our study is our inability to completely rule out the potential impact of the other phenolic compounds we tested because of unknown uptake and cell permeability, but nevertheless, our results add further evidence to this body of research by demonstrating direct effects of PB on tau pathology.

Critically, unlike the polyphenols, OFA and OFB, which disaggregate tau fibrils, PB is a monophenol with a potentially different MOA. We demonstrate that the anti-aggregation effects of PB on tau are sufficient to block seeding by AD patient-derived tau fibrils, in the cell-based tau biosensor seeding assays, but surprisingly, our data suggest that PB inhibits tau monomer aggregation without any disaggregating effects on existing tau fibrils. Future work to assess any potentially synergistic effects between compounds with different MOA could be of great value, assuming no cytotoxicity when treating with both molecules.

Collectively, our study design presents a new approach to quickly screen for bioactive molecules that prevent tauopathy. By

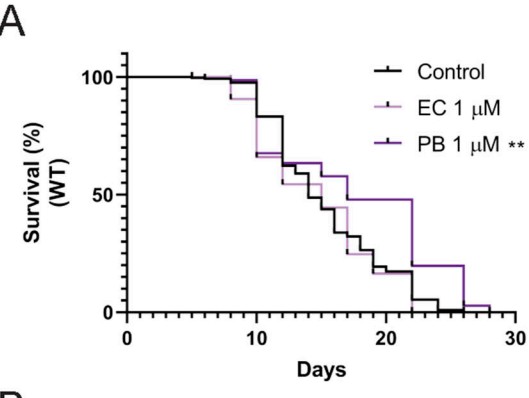

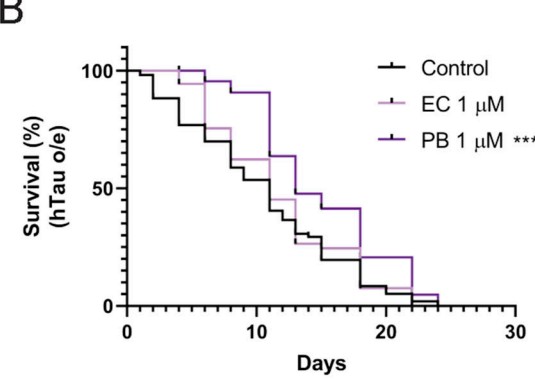

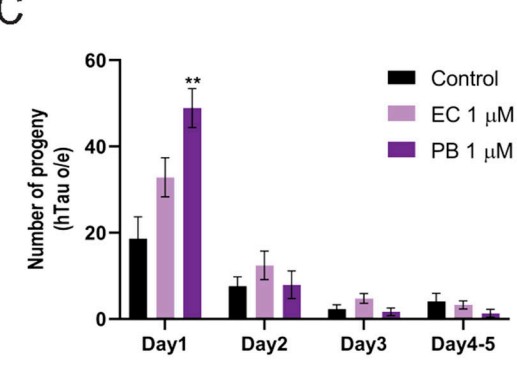

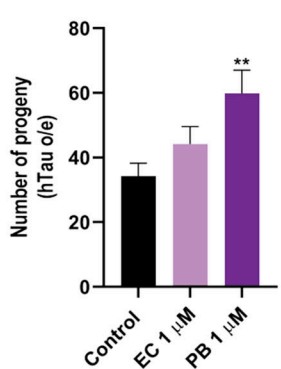

**Figure 5. Effects of echinatin (EC) and plumbagin (PB) on organismal health of *C. elegans*.**
**(A, B)** Survival curves of (A) WT and (B) hTau-expressing (hTau o/e) animals treated with EC and PB. **(C, D)** Effects of EC and PB on daily (C) and total (D)

leveraging two complementary tauopathy models, we reveal natural products can act as potent inhibitors of neuro-degeneration hallmarked by tau protein aggregation. The strength of the approach is demonstrated by revealing inhibitors with very different chemical characteristics and mechanisms of action. Even through limited compound screening, our results show broad phenotypic benefits in models of tau neurodegeneration. Our results add to a growing body of evidence that brain-permeable phenols could be well suited as interventions to counteract systemic proteinopathies and illustrate experimental paradigms for further discovery and development.

## Materials and Methods

### *C. elegans* strains and maintenance

All strains were cultured on nematode growth media (NGM) supplemented with *Escherichia coli* OP50 using standard methods (36). Worms were maintained at 20°C. Strains used in this study include N2 Bristol, referred throughout as WT, and hTau-expressing (hTau o/e) KAE112 (*seals201 [myo-3p::human tau (0N4R;V337M):: unc-54 3'UTR + vha-6p::mCherry::unc-54 3'UTR]*) (9). Age-synchronized populations of worms were obtained by hypochlorite treatment (37).

### Natural product and antioxidant screening

Oolonghomobisflavan A (OFA) (CAS No. 126737-60-8, Cat No. NS240102) and oolonghomobisflavan B (OFB) (CAS No. 176107-91-8, Cat No. NS240202) were purchased from Nagara Science Co.

Ascorbic acid (VC) (CAS No. 50-81-7, Cat No. 95209), caffeic acid (CA) (CAS No. 331-39-5, Cat No. C0625), N-acetylcysteine (NAC) (CAS No. 619-91-1, Cat No. A7250), and plumbagin from *Plumbago indica* (PB) (CAS No. 481-42-5, Cat No. P7262) were purchased from Sigma-Aldrich. Echinatin (EC) (CAS No. 34221-41-5, Cat No. HY-N0269) was purchased from MedChem Express.

### Lifespan assay

Worms were synchronized to generate a synchronous L1 population. Larval stage 4 (L4) worms (identified based on vulval morphology) were moved to NGM agar plates supplemented with M9 buffer (mock treatment control) or OFs including 10 μM OFA, 10 μM OFB, or 5 μM OFA and 5 μM OFB (OFAB). The different concentrations of OFs were prepared in M9 buffer and placed above *E. coli* OP50 lawn and incubated at RT overnight before use. Animals were observed and moved to fresh medium every other day until the end of life. Worms that failed to respond to a gentle touch were scored as dead. Animals with internally hatched

reproductive output. *P < 0.05, **P < 0.01, ***P < 0.001, and ****P < 0.0001, compared with the mock-treated controls by one-way ANOVA followed by Bonferroni's method (post hoc). Worms were maintained at 20°C. n ≥ 30; N = 3. Data represent the mean ± SEM from at least three independent biological replicates.

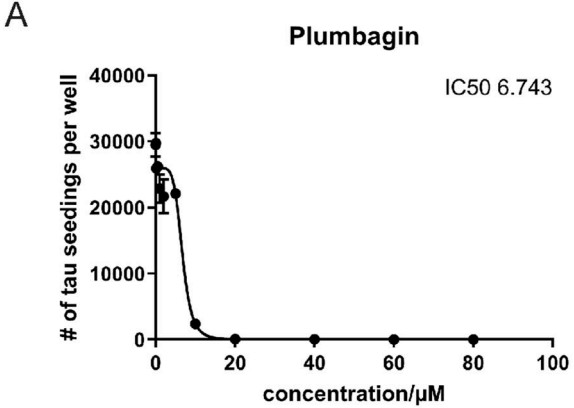

## A

**Plumbagin**

IC50 6.743

## B

***In vitro* seeding assay**

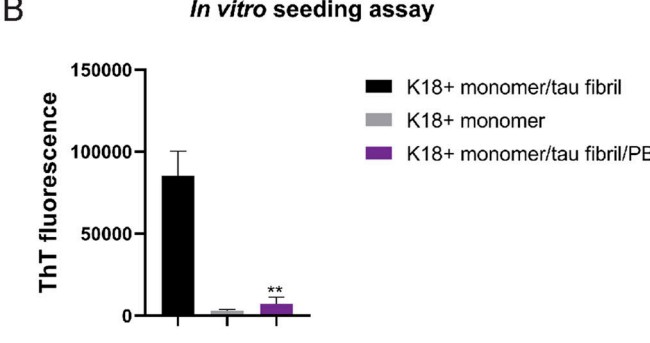

- K18+ monomer/tau fibril
- K18+ monomer
- K18+ monomer/tau fibril/PB

## C

**qEM**

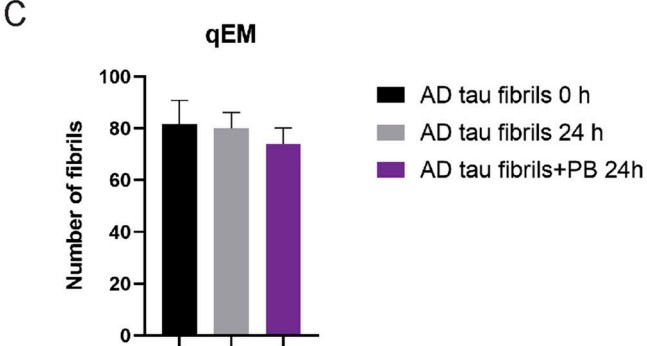

- AD tau fibrils 0 h
- AD tau fibrils 24 h
- AD tau fibrils+PB 24h

## D

**Heparin-induced tau aggregation assay**

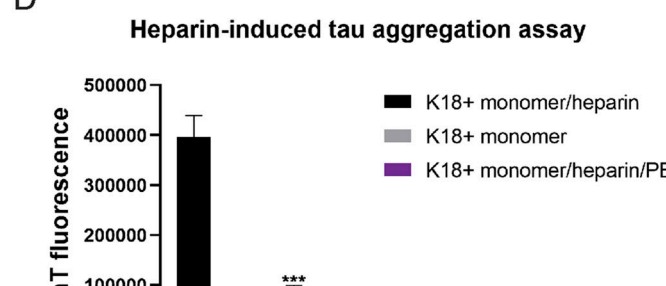

- K18+ monomer/heparin
- K18+ monomer
- K18+ monomer/heparin/PB

**Figure 6. Effects of plumbagin (PB) on AD brain–derived tau fibrils.**
PB is an inhibitor of tau monomer aggregation. **(A)** Dose-dependent inhibition of tau seeding by PB in tau biosensor cell assays using AD crude brain

progeny and extruded gonads, or those that crawled to the side of the plate were censored. Each experimental replicate measured a minimum of 30 individual animals for a total of 90–120 animals/treatment.

### Pharyngeal pumping assay

Pharyngeal pumping assays and lifespan assays were conducted at the same time, specifically on the 5th, 7th, 9th, and 11th d of adulthood of WT and hTau-expressing (hTau o/e) worms. The pharyngeal pumping rates were quantified by counting pharynx contractions for 60 s. Each experimental replicate measured a minimum of 20 individual animals for a total of 60–90 animals/treatment.

### Reproduction assays

WT and hTau-expressing (hTau o/e) worms were synchronized in the same way as in the lifespan assay. The L4 larval stage animals were sorted and placed one by one on each NGM agar plate supplemented with OFs. For brood size assays, L4 worms were singled on a NGM agar plate supplemented with natural products (as indicated) and incubated at 20°C for 24 h. Each group had a minimum of 20 worms. The adult worms were moved every 24 h until egg laying ceased. The eggs were counted using a dissecting microscope every day for 5 d to obtain a number of progeny and a mean brood size.

### WormLab measurement

WT and hTau-expressing (hTau o/e) worms were synchronized by hypochlorite treatment. Eggs were allowed to hatch overnight for a synchronous L1 population on NGM agar plates supplemented with OFs. Worms were then allowed to grow until day 2 and day 4 of adulthood (day 2: 156 h from egg synchronization of WT, 204 h from egg synchronization of hTau o/e; day 4: 204 h from egg synchronization of WT, 252 h from egg synchronization of hTau o/e).

At each time point, worms were washed with M9 buffer (+0.1% Triton) and dropped on an unseeded NGM plate. Worms were allowed to roam for 1 h before recording crawling and thrashing with the MBF Bioscience WormLab microscope. The worms that moved at least 90% of the time were used to analyze with WormLab version 2022 software. To account for differences in worm size, movement speed was normalized by dividing the average speed ($\mu$m/s) by the average body length ($\mu$m).

homogenate as the seed (A). **(B)** In vitro seeding assay with purified tau K18+ monomer and AD tau fibrils as the seed, with aggregation assessed by endpoint ThT fluorescence (B). **(C)** Tau fibril disaggregation evaluated by qEM, showing no apparent disaggregation by PB (C). **(C, D)** Inhibition of tau monomer aggregation by PB, measured as in (C) but without the addition of AD tau fibrils as a seed (D). *$P < 0.05$, **$P < 0.01$, ***$P < 0.001$, and ****$P < 0.0001$, compared with the controls by one-way ANOVA followed by Bonferroni's method (post hoc). N ≥ 60 images. Data represent the mean ± SEM from at least three independent biological replicates.

## Development assay

For body length measurements, WT and hTau-expressing (hTau o/e) worms were synchronized and treated in the same way as described above. Worms were then allowed to grow until each time point (60, 108, and 156 h) and imaged by a MBF Bioscience WormLab microscope. Body length measurements were quantified using WormLab version 2022 software.

## Paralysis assay

hTau-expressing (hTau o/e) worms were synchronized in the same way as in the lifespan assay and treated with OFs at the L4 stage on a NGM agar plate. The number of paralyzed worms was counted from day 1 of adulthood. Worms were classified as paralyzed when they did not move or only moved their head (cleared bacteria giving a halo appearance around the worms' heads). Paralyzed worms were recorded and excluded from the plates every other day.

## RNA sequencing

The L1 larval stage animals were treated with 10 $\mu$M OFA, 10 $\mu$M OFB, or M9 buffer (untreated control). After 48 h from treatment, L4 animals were washed with M9 buffer and frozen in TRI reagent at –80°C until use. Animals were homogenized, and RNA extraction was conducted using the Zymo Direct-zol RNA Miniprep kit (Cat No. R2052). The Qubit RNA BR Assay kit was used to determine RNA concentration. The RNA samples were sent to Novogene to perform RNA sequencing. Read counts were used for differential expression (DE) analysis using the R package DESeq2 (R version 3.5.2). Differentially expressed genes were analyzed using $P$-value < 0.05 and fold change > 1.5 as cutoff.

## Western blot analysis

Synchronized populations of hTau-expressing (hTau o/e) worms were grown to the 3rd d of adulthood. Worms were washed off plates with M9 buffer and fractured by freeze–thaw cycles in liquid nitrogen. The fractured worm biomass was ground and lysed in FA buffer (1 mM EDTA, pH 8.0, 0.1% wt/vol sodium deoxycholate, 1% vol/vol Triton X-100, 1x HALT protease inhibitor). Total protein concentrations were quantified by the Bradford assay (Sigma-Aldrich). An equal amount of protein (20 $\mu$g) was separated on 4–12% Bis-Tris polyacrylamide gel (Invitrogen) in MOPS running buffer (Invitrogen) and then transferred to nitrocellulose membranes (GE Healthcare Life Sciences). After blocking for 1 h with 3% BSA in PBST (PBS, 0.1% Tween-20), the membranes were subjected to immunoblot analysis. Antibodies used include the following: pTau S202 clone D4H7E (1:1,000; Cell Signaling), pTau S416 clone D7U2P (1:1,000; Cell Signaling), pan-tau (1:1,000; Millipore Sigma), histone H3 (1:5,000; Abcam), and HRP-conjugated secondary antibodies (1:10,000; Thermo Fisher Scientific). Specific protein bands were visualized and evaluated using FluorChem HD2 (ProteinSimple).

## Statistical analysis

Data are presented as the mean ± SEM (n, indicated for each experiment, replicated a minimum of three times). Data were analyzed by one-way ANOVA followed by Bonferroni's method (post hoc). Data handling and statistical processing were performed using GraphPad Prism 9.0. Differences were considered significant at the $P \leq 0.05$ level.

## K18CY cell culture

HEK293T cell lines that stably express tau-K18CY labeled with green fluorescent protein (GFP) obtained from Marc Diamond's laboratory at the University of Texas Southwestern Medical Center (38) were used. The cells were cultured in a T25 flask in DMEM (Cat No. 11965092; Life Technologies) supplemented with 10% (vol/vol) FBS (Cat No. A3160401; Life Technologies), 1% penicillin/streptomycin (Cat No. 15140122; Life Technologies), and 1% GlutaMAX (Cat No. 35050061; Life Technologies) at 37°C and 5% $CO_2$ in a humidified incubator. To test the inhibitors on the biosensor cells, 100 $\mu$l of cells was plated in 96-well plates and stored in the 37°C, 5% $CO_2$ incubator for 16–24 h before transfection.

## Biosensor cell seeding assays

EGCG (control) and OFs were diluted in DMSO to 1.4 mM stocks. 10 $\mu$M of OFs and 100 $\mu$M of PB were used as concentration treatment. Homogenized AD brain was diluted in Opti-MEM (Cat No. 31985062; Thermo Fisher Scientific) in a 1:20 ratio. The diluted brain homogenate was incubated with indicated inhibitors for 16–24 h at 4°C. Inhibitor-treated seeds were sonicated again in a Cup Horn (Qsonica, MPH) water bath for 3 min at 40% power and then mixed with a 1–20 solution of Lipofectamine 2000 (Cat No. 11668019; Thermo Fisher Scientific) and Opti-MEM. The Lipofectamine creates a liposome around the fibrils to allow delivery into the cells. After 20 min, 10 $\mu$l of inhibitor-treated fibrils was added to the previously plated 100 $\mu$l of cells in triplicate, avoiding use of the perimeter wells to yield a final ligand concentration of 10 mM on cells. In vitro tau aggregation was performed by incubating recombinant tau monomers with heparin, a polyanionic cofactor that induces fibrillization. This method is distinct from seeding assays using AD-derived tau fibrils, which promote aggregation through a prion-like templating mechanism.

## Preparation of crude Alzheimer's brain–derived tau seeds

Human Alzheimer's brain autopsy samples were obtained from the University of California, Los Angeles (UCLA) Pathology Department according to US Department of Health and Human Services regulations from patients who consented to autopsy. Approximately 0.2$g$ of tissue was excised, and a Kinematica PT 10-35 POLYTRON was used to homogenize the tissue with 0.75 ml sucrose buffer (0.8 M NaCl, 10% sucrose, 10 mM Tris–HCl, pH 7.4) supplemented with 1 mM EGTA at levels 4–5 in 15-ml Falcon tubes. Homogenates were aliquoted and used for seeding as described previously. For qEM studies, tau PHFs were further purified from homogenates by

sarkosyl extraction. Briefly, 0.5–1.0$g$ homogenized tissue was centrifuged at 15,300 rpm for 20 min. The supernatant was adjusted to a final concentration of 1% sarkosyl and incubated for 1 h at RT with shaking at 250 rpm. Fibrils were obtained by ultracentrifugation at 95,000 rpm (403,600$g$) for 1 h. Pellets were resuspended in sucrose buffer supplemented with 1 mM EGTA and 5 mM ethylenediaminetetraacetic acid (EDTA) and centrifuged once more at 15,300 rpm for 20 min followed (for the supernatant) by ultracentrifugation at 95,000 rpm for 1 h. The final pellet was resuspended in 0.1 ml of 20 mM Tris–HCl, pH 7.4, 100 mM NaCl.

### Negative stain grid preparation

Purified Alzheimer's brain–derived tau PHF fibrils were diluted 1:10 in PBS and incubated with OFA for indicated time points at 4°C. For qEM, after 48-h incubation, EM grids were prepared by depositing 6 $\mu$l of samples on formvar/carbon-coated copper grids (400 mesh) for 3 min with inhibitor preincubation times of either 0 h (negative control) or 48 h (positive control). The sample was rapidly and carefully removed by fast blot using filter paper without drying the grid and stained with 4% uranyl acetate for 2 min, then wicked dry by filter paper. Automated images were collected using the FEI Glacios driven by EPU software. Visible fibrils were counted from 66 images, each for the 0- and 24-h OFA incubation time points, and fibrils were plotted by dividing counted images into thirds to evaluate the standard error.

## Data and Materials Availability

All data are available in the main text or the supplementary materials.

## Supplementary Information

## Acknowledgements

We thank S Keel and NT Phan for technical assistance and CM Ramos for critical reading of the manuscript. This work was funded by the NIH R01AG058610 and Hevolution Foundation Award HF AGE-004 to SP Curran, an AFAR Postdoctoral fellowship to CD, and a pilot grant to PM Seidler that is funded in part by the Nathan Shock Center of Excellence P30AG068345. We also thank the USC School of Gerontology Imaging Core. Some strains were provided by the CGC, which is funded by the NIH Office of Research Infrastructure Programs (P40 OD010440). We thank WormBase for database curation and data access.

### Author Contributions

CD Shepard: formal analysis, investigation, visualization, and writing—original draft.
X Chang: investigation.
PM Seidler: investigation.
SP Curran: conceptualization, formal analysis, supervision, funding acquisition, investigation, visualization, project administration, and writing—original draft, review, and editing.

### Conflict of Interest Statement

The authors declare that they have no conflict of interest.

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
