## [Reviewer comments · Life Science Alliance]

Polyphyletic screen defines distinct classes of plant-derived natural products that oppose tauopathy

Chatrawee Shepard, Xinmin Chang, Paul Seidler, and Sean Curran

DOI: <https://doi.org/10.26508/lsa.202503393>

Corresponding author(s): Sean Curran, University of Southern California and Chatrawee Shepard, University of Southern California

Review Timeline:

Submission Date:	2025-05-20
Editorial Decision:	2025-07-14
Revision Received:	2025-10-04
Editorial Decision:	2025-10-27
Revision Received:	2025-10-31
Accepted:	2025-11-04

Scientific Editor: Tim Fessenden

Transaction Report:

July 14, 2025

Re: Life Science Alliance manuscript #LSA-2025-03393-T

Prof. Sean P Curran
University of Southern California
Leonard Davis School of Gerontology
3715 McClintock Avenue
Los Angeles, CA 90089-0191

Dear Dr. Curran,

Thank you for submitting your manuscript entitled "Polyphyletic screen defines distinct classes of plant-derived natural products that oppose tauopathy". The manuscript has been evaluated by expert reviewers, whose reports are appended below. We appreciate your patience during the unusually long review period. Unfortunately, after an assessment of the reviewer feedback, our editorial decision is against publication in Life Science Alliance at this time.

As you will see, reviewers appreciated the novel observations on effects of OFA/OFB, as well as novel natural compounds, on tau-mediated toxicity. However, both reviewers expressed significant concerns over validation and controls of key observations and the use and interpretation of the lifespan extension reported here. Namely, Reviewer 1 remarked that essential controls to delineate promoter activity from protein levels, and to distinguish total vs phosphorylated tau, were missing. Reviewer 2 noted that effects on lifespan may result from complex effects of tau aggregates not formally tested, nor properly acknowledged, in this work. This reviewer further noted the absence of an initial mortality-free plateau rendered by the tested compounds, which at a minimum must be acknowledged in the text. These concerns, which relate to the central advances set forth in this work, prevent further consideration of this manuscript in its current form.

Although your manuscript is intriguing, the points raised by the reviewers are more substantial than can be addressed in a typical revision period. If you wish to expedite publication of the current data, it may be best to pursue publication at another journal.

Given the interest in the topic, I would be open to re-submission to Life Science Alliance of a significantly revised and extended manuscript that fully addresses all reviewers' concerns and is subject to further peer review. If you would like to resubmit this work to Life Science Alliance, you may submit an appeal directly through our manuscript submission system. An appeal request must be accompanied by a revision plan with a point-by-point rebuttal.

Regardless of how you choose to proceed, we hope that the comments below will prove constructive as your work progresses.

Thank you for thinking of Life Science Alliance as an appropriate place to publish your work.

Sincerely,

Reviewer #1 (Comments to the Authors (Required)):

In previous work, this group demonstrated that plant-derived natural products, Olonghomobisflavin A (OFA) and Olonghomobisflavin B (OFB), extend the lifespan of *C. elegans* and reduce A β - and polyQ-induced neuro/proteotoxicity, representing models of Alzheimer's and Parkinson's disease, respectively (Duangjan and Curran, 2021, Gerontology). In the present manuscript, the authors extend these findings by evaluating OFA and OFB in *C. elegans* expressing human Tau (hTau), an additional model for Alzheimer's disease. They show that OFA and OFB treatments enhance both lifespan and muscle function in hTau-expressing worms. Furthermore, they demonstrate that these compounds ameliorate fibril accumulation in mammalian cells seeded with brain homogenates from Alzheimer's disease patients, supporting the potential of OFA and OFB to reduce proteotoxicity. Additionally, the authors identify plumbagin as a potent inhibitor of Tau aggregation. This manuscript presents novel and potentially impactful findings that extend prior work on plant-derived compounds in neurodegeneration models, and contributes valuable insights into potential therapeutic approaches for neurodegenerative diseases using plant-derived natural products. However, the paper requires substantial revisions in data presentation, figure labeling, and textual

clarity.

Comments on figures

Figure 1:

Including chemical structures of the compounds would be more informative.

The label "Control vs. OFA" is unclear; rephrasing to "OFA vs. Control" would better indicate the treatment comparison.

Highlighting genes commonly up- or downregulated by both OFA and OFB treatments (e.g., through a Venn diagram) would be helpful.

Font size in panels D-H is too small and should be increased for readability.

Figure 2:

"hTau-expressing worms" is more descriptive than the strain name "KAE112", which lacks context.

The unit for body length should be micrometers (μm), not micromolar (μM). If μm is correct, the measured lengths seem unusually short for adult worms (normally ~ 1 mm); this discrepancy should be addressed.

Normalization methods for movement speed in panels E and F should be described.

In panel I, the y-axis tick for 150% is unnecessary since the graph shows percentages and does not exceed 100%.

Figure 4:

Clarify whether the reduced tau phosphorylation shown in panel A is due to decreased total Tau protein or decreased phosphorylation levels.

In panel D, the gel is labeled as "total Tau," but described as showing Tau aggregates in the text. This contradiction needs to be resolved.

It should also be explained why Tau aggregates appear at specific sizes on the gel, and the kDa labels are missing.

Since the Tau transgene is driven by the myo-3 promoter, it is important to distinguish whether OFA and OFB treatments affect Tau expression (e.g., the activity of the myo-3 promoter) or its aggregation. The current data do not clearly differentiate between the two possibilities.

The same gel image appears in both Figure 4D and Figure S4A. Duplicate use of data across figures must be justified or corrected.

In panels H and I, replacing "OFA" and "OFB" with "AD + OFA" and "AD + OFB" would clarify the experimental conditions.

Figure 6:

The number of Tau aggregates per well at 0 μM in Figure 6A is ~ 5 times higher than in Figure S6A; this inconsistency needs explanation.

Data points between 0 and 20 μM concentrations are difficult to distinguish, and concentrations higher than 20 μM might not be necessary.

Y-axis labels between Figures 6A and S6A differ for what appears to be the same experiment. This should be standardized.

In panels 6B and 6D, the data legends do not match the x-axis labels and are confusing; since only three conditions are shown, only three labels should be used.

The term "heparin-induced tau aggregation" is used without explanation and should be defined in the text.

Comments on Text

There are missing spaces in the citations in the first paragraph of the Introduction.

Line 81: Consider using just "NFTs", as the abbreviation was already introduced in line 67.

Line 108: The sentence begins with "In previous work, we identified..." but also cites other studies [14-16].

Line 120: "treatment" should be "treatments".

Line 122: Define WT as "wild-type (WT)" upon first use.

Line 152: "The ability OFs..." should be "The ability of OFs...".

Line 160: Reference to "Figure S3A" should be "Figure S2A."

Line 161: The claim that OFA and OFB do not affect slowed growth is ambiguous. Clarify whether hTau worms grow more slowly or simply reach a smaller final size compared to N2.

Line 170: "Pharyngeal muscle paralysis" is misleading without evidence; consider rephrasing as "reduced pharyngeal pumping."

Lines 164-180: The same pharyngeal data are discussed in the two paragraphs. Consider merging for clarity and flow.

Line 188: In line 188, "synergistic" may be "additive" because it would have an additive effect if the two treatments are independent. Synergistic indicates that two treatments have an interaction.

Line 196: The subtitle is italicized.

Line 199: "Hereafter called hTau-expressing" should be rephrased, as the term was used earlier.

Line 246: They say "PB, a natural product derived from *Plumbago zeylanica*, significantly increased 246 the lifespan of both h-Tau expressing tg and wildtype strains, although ascorbic acid (VC), caffeic acid (CA), echinatin (EC) and N-acetylcysteine (NAC) had no effect on lifespan (Figure 5A-B)," There is no data of VC, CA and NAC in Figure 5A-B. This should be revised. Additionally, "tg" in the sentence needs clarification.

Line 252: Specify figure panel in reference to "Figure S5."

Line 256: Use "PB" instead of "plumbagin" for consistency with earlier abbreviation.

Line 265: Consider removing "MOA" unless the abbreviation is used elsewhere in the section. Also, "Mechanism of action (MOA)" is used in line 313.

Line 289: The result on heat shock protein expression should be described in the Results section.
Line 313: Briefly explain what MARK4 is for clarity.
Line 373: "Worms" is repeated in the sentence.
Line 449: The method used to sort L4 worms should be clarified.
Line 451: Specify what "oolong tea extracts" are.
Line 514: Citation formatting is not consistent.

Reviewer #2 (Comments to the Authors (Required)):

Summary

This study presents credible comparisons of RNA expression shifts induced by exposure to two natural products (constituents of oolong tea), Oolonghomobisflavin A (OFA) and Oolonghomobisflavan B (OFB). Annotation-term enrichment (a.k.a. GO meta-analysis) implicated relief of stresses (especially oxidative), and pathways associated with proteostasis and longevity. The transcriptomic analyses constitute the strongest components of this paper. The authors then assessed their ability to "rescue" several tauopathic traits exhibited by a *C. elegans* strain expressing a human tau transgene. Next, they "performed screening with several polyphenols with documented antioxidative effects. These included ascorbic acid (VC), caffeic acid (CA), echinatin (EC), N-acetylcysteine (NAC), and plumbagin (PB)."

Critique

Please provide a better rationale for employing a lifespan survival assay as "a new strategy for identifying therapeutic compounds that target tauopathy". In our experience, well-conducted longevity survivals are more painstaking and time consuming than tau aggregation or disaggregation assays, and most critically, they can reflect the effects of many pathways other than tauopathy. Absent this rationale, the present paper does not qualify as an advance in methodology per se. Please note that while plumbagin and echinatin are diphenols, caffeic acid is monophenolic, and ascorbic acid and NAC are not phenolic at all. To describe this set of compounds as "several polyphenols" is therefore misleading. Plumbagin was observed to extend lifespan of a tau-expressing *C. elegans* strain, but failed to "rectangularize" it; i.e., the PB plots did not acquire an initial mortality-free plateau. While not quite essential, restoration of such a plateau would make a better case for lifespan rescue by the assessed drugs.

Specific points to improve upon:

The tau-expressing strain should be clearly presented in Results, to describe the nature of the two human AD-associated mutations and to emphasize that tau expression under myo-3p is chiefly limited to muscle, whereas vha-6p directs mCherry expression to the intestine. These are important aspects of the experimental design that should not be taken for granted.

Figure 1: the smallest font is illegible: even with max. enlargement, I cannot read transcript names (y axes) or x axis labels in panels D - H.

Figure 3: Need to state the temperature survivals were conducted at! This information is in Methods, but needs to be stated in Results as well. OFA/B both fail to rectangularize survival curves of tau-expressing worms, which is a bit worrying.

Figure 4 is difficult to interpret! It would be helpful to enlarge panels K - N, and perhaps replace the arrows with like-colored circles to better indicate what is meant to be marked.

Figure 5: It is not possible to distinguish the survival plots based on the keys provided (panels A and B). It appears that these survivals may have been conducted with insufficient N.

REFEREE CROSS-COMMENTS

I concur with the detailed comments of Reviewer 1, and if it is his/her opinion that this paper (once revised) does qualify for publication in Life Science Alliance, I am happy to concede that point. In any case, substantial revision will be necessary to address all of the concerns of both reviewers.

We thank the reviewers for their insight to improve the readability of our manuscript. We apologize that in some sections the data and methods were not as clear as they could have been, but we have edited to ensure that all concerns were addressed. We provide a point-by-point response to each concern raised by each reviewer.

Reviewer #1 (Comments to the Authors (Required)):

In previous work, this group demonstrated that plant-derived natural products, Olonghomobisflavin A (OFA) and Olonghomobisflavin B (OFB), extend the lifespan of *C. elegans* and reduce A β - and polyQ-induced neuro/proteotoxicity, representing models of Alzheimer's and Parkinson's disease, respectively (Duangjan and Curran, 2021, Gerontology). In the present manuscript, the authors extend these findings by evaluating OFA and OFB in *C. elegans* expressing human Tau (hTau), an additional model for Alzheimer's disease. They show that OFA and OFB treatments enhance both lifespan and muscle function in hTau-expressing worms. Furthermore, they demonstrate that these compounds ameliorate fibril accumulation in mammalian cells seeded with brain homogenates from Alzheimer's disease patients, supporting the potential of OFA and OFB to reduce proteotoxicity. Additionally, the authors identify plumbagin as a potent inhibitor of Tau aggregation. This manuscript presents novel and potentially impactful findings that extend prior work on plant-derived compounds in neurodegeneration models, and contributes valuable insights into potential therapeutic approaches for neurodegenerative diseases using plant-derived natural products. However, the paper requires substantial revisions in data presentation, figure labeling, and textual clarity.

Comments on figures

Figure 1:

Including chemical structures of the compounds would be more informative.

RESPONSE: We have included the structure of OFA and OFB compounds in Figure 1.

The label "Control vs. OFA" is unclear; rephrasing to "OFA vs. Control" would better indicate the treatment comparison.

RESPONSE: We apologize for the confusion; the volcano plots show gene expression differences between the treatment group (OFA or OFB) and the mock-treated control group; the data are the change in the treatment group relative to the control. These results are consistent with the full dataset presented in Table S1.

Highlighting genes commonly up- or downregulated by both OFA and OFB treatments (e.g., through a Venn diagram) would be helpful.

RESPONSE: We have addressed this by including a full list of the genes altered by treatment (OFA or OFB) compared to mock-treated controls is presented in the Table S1; we feel this higher level of resolution is more useful to readers than a Venn diagram. We have also included the top five

biologically relevant genes that were most significantly altered by the treatment (Figure 1A-B, Figure S1A).

Font size in panels D-H is too small and should be increased for readability.

RESPONSE: We have increased the font size as requested.

Figure 2:

"hTau-expressing worms" is more descriptive than the strain name "KAE112", which lacks context.

We have changed labels to "hTau o/e" for human Tau overexpression. KAE112 is only left in the list of strains used in the Methods section.

The unit for body length should be micrometers (μm), not micromolar (μM). If μm is correct, the measured lengths seem unusually short for adult worms (normally $\sim 1\text{ mm}$); this discrepancy should be addressed.

RESPONSE: We apologize for this typo which we have changed μM to μm . The reviewer is correct that adults worms are normally $\sim 1\text{mm}$, which is what is documented in the figures; $>800\ \mu\text{m}$ (0.8mm) at 60 hours post-hatching and $>1100\ \mu\text{m}$ (1.1mm) at 156 hours post-hatching.

Normalization methods for movement speed in panels E and F should be described.

RESPONSE: We have added more detail in method section (line 481).

In panel I, the y-axis tick for 150% is unnecessary since the graph shows percentages and does not exceed 100%.

RESPONSE: We fixed this in the revised version.

Figure 4:

Clarify whether the reduced tau phosphorylation shown in panel A is due to decreased total Tau protein or decreased phosphorylation levels.

RESPONSE: In this revised version (line 207), we normalized phospho-tau levels to total tau. We found that the pTau/total tau ratio was reduced in treated animals as compared to mock-treated controls at both Ser202 (OFA=67.56 \pm 0.06%, OFB 88.07 \pm 0.07%) and Ser416 (OFA=87.58 \pm 0.04% OFB=75.17 \pm 0.01), suggesting even though total tau is reduced, the reduction on tau phosphorylation is significantly reduced as well.

In panel D, the gel is labeled as "total Tau," but described as showing Tau aggregates in the text. This contradiction needs to be resolved.

RESPONSE: We have revised the text to clarify this point (line 203).

It should also be explained why Tau aggregates appear at specific sizes on the gel, and the kDa labels are missing.

RESPONSE: We have added the molecular weight markers as requested to this panel.

Since the Tau transgene is driven by the *myo-3* promoter, it is important to distinguish whether OFA and OFB treatments affect Tau expression (e.g., the activity of the *myo-3* promoter) or its aggregation. The current data do not clearly differentiate between the two possibilities.

The Tau transgene is driven by the *myo-3* promoter, we examined RNA-seq data from N2 worms treated with OFA or OFB and observed downregulation of *myo-3* (K12F2.1), suggesting that reduced **overall** levels of Tau protein from the transgene may partially be due to decreased transgene expression (line 208).

This is consistent with the total Tau protein levels that we previously reported in Figure 4 in the first submission following OFA and OFB treatment.

As stated above, we normalize the measured phospho-Tau levels to total Tau and found that the resulting pTau/total Tau ratios were significantly reduced for both Ser202 and Ser416 (line 206).

These findings indicate that even after adjusting for reduced Tau expression, phosphorylation levels remain significantly decreased, suggesting that OFA and OFB directly affect Tau phosphorylation and possibly aggregation.

The same gel image appears in both Figure 4D and Figure S4A. Duplicate use of data across figures must be justified or corrected.

RESPONSE: We apologize for this oversight and removed S4A.

In panels H and I, replacing "OFA" and "OFB" with "AD + OFA" and "AD + OFB" would clarify the experimental conditions.

RESPONSE: We have edited as suggested.

Figure 6:

The number of Tau aggregates per well at 0 μM in Figure 6A is ~ 5 times higher than in Figure S6A; this inconsistency needs explanation.

RESPONSE: The discrepancy arises because different AD brain-derived tau seeds were used in Figures 6A and S6A. The level of tau aggregation can vary significantly depending on several biological factors, including disease stage, brain region, and patient age. These intrinsic differences in seed potency highlight the importance of including internal controls within each experiment, allowing for reliable comparisons of treatment effects within the same experimental context; this is outlined in the methods section (line 538).

Data points between 0 and 20 μM concentrations are difficult to distinguish, and concentrations higher than 20 μM might not be necessary.

RESPONSE: We agree that the curves between 0 and 10 μM are closely spaced. However, a noticeable drop in tau aggregation is observed at 10 μM , which closely approaches the effect seen at 20 μM . Since 10 μM is a biologically relevant and commonly used concentration for evaluating compound efficacy, we believe it sufficiently demonstrates the inhibitory effect. Higher concentrations were included to confirm dose-response behavior.

Y-axis labels between Figures 6A and S6A differ for what appears to be the same experiment. This should be standardized. In panels 6B and 6D, the data legends do not match the x-axis labels and are confusing; since only three conditions are shown, only three labels should be used.

RESPONSE: We have revised as suggested.

The term "heparin-induced tau aggregation" is used without explanation and should be defined in the text.

RESPONSE: "heparin-induced tau aggregation" refers to an *in vitro* method where recombinant tau monomers are aggregated into fibrils using heparin as a polyanionic inducer. This contrasts with tau seeding using brain-derived fibrils, where preformed pathological fibrils extracted from AD brain tissue are used to seed aggregation of tau monomers. We have included these details in the method section of revised version (line 548).

Comments on Text (as also indicated below, we have implemented all of the reviewer's suggestions in the revised version.)

There are missing spaces in the citations in the first paragraph of the Introduction.

Line 81: Consider using just "NFTs", as the abbreviation was already introduced in line 67.

Line 108: The sentence begins with "In previous work, we identified..." but also cites other studies [14-16].

Line 120: "treatment" should be "treatments".

Line 122: Define WT as "wild-type (WT)" upon first use.

Line 152: "The ability OFs..." should be "The ability of OFs...".

Line 160: Reference to "Figure S3A" should be "Figure S2A."

Line 161: The claim that OFA and OFB do not affect slowed growth is ambiguous. Clarify whether hTau worms grow more slowly or simply reach a smaller final size compared to N2.

Line 170: "Pharyngeal muscle paralysis" is misleading without evidence; consider rephrasing as "reduced pharyngeal pumping."

Lines 164-180: The same pharyngeal data are discussed in the two paragraphs. Consider merging for clarity and flow.

Line 188: In line 188, "synergistic" may be "additive" because it would have an additive effect if the two treatments are independent. Synergistic indicates that two treatments have an interaction.

Line 196: The subtitle is italicized.

Line 199: "Hereafter called hTau-expressing" should be rephrased, as the term was used earlier.

Line 246: They say "PB, a natural product derived from *Plumbago zeylanica*, significantly increased 246 the lifespan of both h-Tau expressing tg and wildtype strains, although ascorbic acid (VC), caffeic acid (CA), echinatin (EC) and N-acetylcysteine (NAC) had no effect on lifespan (Figure 5A-B)," There is no data of VC, CA and NAC in Figure 5A-B. This should be revised. Additionally, "tg" in the sentence needs clarification.

Line 252: Specify figure panel in reference to "Figure S5."

Line 256: Use "PB" instead of "plumbagin" for consistency with earlier abbreviation.

Line 265: Consider removing "MOA" unless the abbreviation is used elsewhere in the section. Also, "Mechanism of action (MOA)" is used in line 313.

Line 289: The result on heat shock protein expression should be described in the Results section.

Line 313: Briefly explain what MARK4 is for clarity.

Line 373: "Worms" is repeated in the sentence.

Line 449: The method used to sort L4 worms should be clarified.

Line 451: Specify what "oolong tea extracts" are.

Line 514: Citation formatting is not consistent.

RESPONSE: We have implemented all of the reviewer's suggestions in the revised version.

Reviewer #2 (Comments to the Authors (Required)):

Summary

This study presents credible comparisons of RNA expression shifts induced by exposure to two natural products (constituents of oolong tea), Oolonghomobisflavin A (OFA) and Oolonghomobisflavan B (OFB). Annotation-term enrichment (a.k.a. GO meta-analysis) implicated relief of stresses (especially oxidative), and pathways associated with proteostasis and longevity. The transcriptomic analyses constitute the strongest components of this paper. The authors then assessed their ability to "rescue" several tauopathic traits exhibited by a *C. elegans* strain expressing a human tau transgene. Next, they "performed screening with several polyphenols with documented antioxidative effects. These included ascorbic acid (VC), caffeic acid (CA), echinatin (EC), N-acetylcysteine (NAC), and plumbagin (PB)."

Critique

Please provide a better rationale for employing a lifespan survival assay as "a new strategy for identifying therapeutic compounds that target tauopathy". In our experience, well-conducted longevity survivals are more painstaking and time consuming than tau aggregation or disaggregation assays, and most critically, they can reflect the effects of many pathways other than tauopathy. Absent this rationale, the present paper does not qualify as an advance in methodology per se.

RESPONSE: While lifespan survival assays do take 2-3 weeks to complete, their inclusion in our screening method is intentional and strategic. We are interested in improve overall health and the correlation of aggregation and pathology are not always aligned. Lifespan assays in *C. elegans* expressing human tau (hTau o/e) offer a whole-organism perspective on how well OFA and OFB treatment can help maintain protein balance and copes with tau-induced toxicity. We do not propose lifespan assays as a replacement for biochemical tau aggregation/disaggregation assays, but rather as a complementary strategy that can take into account whole animal physiology. We have implemented in the revised version (line 173, line 275).

Please note that while plumbagin and echinatin are diphenols, caffeic acid is monophenolic, and ascorbic acid and NAC are not phenolic at all. To describe this set of compounds as "several polyphenols" is therefore misleading.

RESPONSE: We have changed this to "antioxidants including an additional monophenol and diphenol" (line 328).

Plumbagin was observed to extend lifespan of a tau-expressing *C. elegans* strain, but failed to "rectangularize" it; i.e., the PB plots did not acquire an initial mortality-free plateau. While not quite essential, restoration of such a plateau would make a better case for lifespan rescue by the assessed drugs.

While PB significantly extended median lifespan, but not maximum lifespan, in the hTau o/e worms. This pattern may reflect a partial rescue of tau-induced toxicity, but not a complete normalization of

organismal physiology. We agree that restoring early plateau would strengthen the case. We see this as an important goal for future compound optimization. We have implemented in the revised version (line 253).

Specific points to improve upon:

The tau-expressing strain should be clearly presented in Results, to describe the nature of the two human AD-associated mutations and to emphasize that tau expression under myo-3p is chiefly limited to muscle, whereas vha-6p directs mCherry expression to the intestine. These are important aspects of the experimental design that should not be taken for granted.

RESPONSE: We added and emphasized hTau o/e worms in the results section as suggested (line 206).

Figure 1: the smallest font is illegible: even with max. enlargement, I cannot read transcript names (y axes) or x axis labels in panels D - H.

RESPONSE: We have increased the font as requested.

Figure 3: Need to state the temperature survivals were conducted at! This information is in Methods, but needs to be stated in Results as well. OFA/B both fail to rectangularize survival curves of tau-expressing worms, which is a bit worrying.

RESPONSE: We revised as suggested (line 379, 411). The results suggest that while OFs helps reduced the harmful effects of tau over time, it may not completely reverse the damage (perhaps prior to treatment).

Figure 4 is difficult to interpret! It would be helpful to enlarge panels K - N, and perhaps replace the arrows with like-colored circles to better indicate what is meant to be marked.

RESPONSE: We have moved the arrows to more clearly indicate what is being pointed out. Note we have increased the resolution of the image so that when zooming in digitally you can still see all data. We prefer to keep the entire field rather than enlarge a smaller section. Also note that we tried using circles instead of arrows, but this covers more data in the image that we would prefer readers clearly see.

Figure 5: It is not possible to distinguish the survival plots based on the keys provided (panels A and B). It appears that these survivals may have been conducted with insufficient N.

All lifespan assays (informed by power analysis and decades of experience) were conducted in biological triplicate (30-50 worms/replicate), consisting of approximately 90-155 animals. This was sufficient for measuring differences of statistical significance.

REFEREE CROSS-COMMENTS

I concur with the detailed comments of Reviewer 1, and if it is his/her opinion that this paper (once revised) does qualify for publication in Life Science Alliance, I am happy to concede that point. In any case, substantial revision will be necessary to address all of the concerns of both reviewers.

RESPONSE: We have made all requested modifications and addressed all points raised by the reviewers.

October 27, 2025

RE: Life Science Alliance Manuscript #LSA-2025-03393-TR-A

Prof. Sean P Curran
University of Southern California
Leonard Davis School of Gerontology
3715 McClintock Avenue
Los Angeles, CA 90089-0191

Dear Dr. Curran,

Thank you for submitting your revised manuscript entitled "Polyphyletic screen defines distinct classes of plant-derived natural products that oppose tauopathy". As you will see, reviewers are now satisfied and recommend publication. We would be happy to publish your paper in Life Science Alliance pending final revisions necessary to meet our formatting guidelines.

- Please upload your main and supplementary figures as single files; all figure legends should only appear in the main manuscript file (at the end of the file).
- Please remove Figures from main manuscript file.
- Please add a statement concerning the approval for human autopsy samples. See: <https://www.life-science-alliance.org/editorial-policies#humans>
- Supplementary table legends should appear in manuscript file after the Figure Legends.
- Please add ORCID ID for secondary corresponding author--they should have received instructions on how to do so.
- Please add the X and Bluesky handles of your host institute/organization as well as your own or/and one of the authors in our system.
- The titles in both the system and the manuscript file must be consistent with each other.
- In the reference list, citations should be listed with the authors' surnames and initials inverted. Where there are more than 10 authors on a paper, the first 10 will be listed, followed by 'et al.'
- Please add callouts for Figure S5A-F and J-L to your main manuscript text .

A. FINAL FILES:

B. MANUSCRIPT ORGANIZATION AND FORMATTING:

Thank you for your attention to these final processing requirements. Please revise and format the manuscript and upload materials as soon as you are able.

Sincerely,

Reviewer #1 (Comments to the Authors (Required)):

The authors have adequately addressed my concerns, and the manuscript has been significantly improved.

Referee Cross-Comments

I have no conflicts with the other reviewer's comments.

Reviewer #2 (Comments to the Authors (Required)):

1. Summary

This study presents credible comparisons of RNA expression shifts induced by exposure to two natural products (constituents of oolong tea), Oolonghomobisflavin A (OFA) and Oolonghomobisflavan B (OFB). Annotation-term enrichment (a.k.a. GO meta-analysis) implicated relief of stresses (especially oxidative), and pathways associated with proteostasis and longevity. The transcriptomic analyses constitute the strongest components of this paper. The authors then assessed their ability to "rescue" several tauopathic traits exhibited by a *C. elegans* strain expressing a human tau transgene. Next, they "performed screening with several polyphenols with documented antioxidative effects. These included ascorbic acid (VC), caffeic acid (CA), echinatin (EC), N-acetylcysteine (NAC), and plumbagin (PB)."

2. The manuscript, as revised, is acceptable for publication by LSA.

3. There are no issues outstanding that should deter or delay publication.

November 4, 2025

RE: Life Science Alliance Manuscript #LSA-2025-03393-TRR

Prof. Sean P Curran
University of Southern California
Leonard Davis School of Gerontology
3715 McClintock Avenue
Los Angeles, CA 90089-0191

Dear Dr. Curran,

Thank you for submitting your Research Article entitled "Polyphyletic screen defines distinct classes of plant-derived natural products that oppose tauopathy". It is a pleasure to let you know that your manuscript is now accepted for publication in Life Science Alliance. Congratulations on this interesting work.

Your manuscript will now progress through copyediting and proofing. During proofing we suggest you consider breaking up the third sentence in the abstract for improved readability. It is journal policy that authors provide original data upon request.

DISTRIBUTION OF MATERIALS:

Again, congratulations on a very nice paper. I hope you found the review process to be constructive and are pleased with how the manuscript was handled editorially. We look forward to future exciting submissions from your lab.

Sincerely,
